# Estimating Rainfall Interception of *Pinus hartwegii* and *Abies religiosa* Using Analytical Models and Point Cloud

**Claudia Bolaños-Sánchez** [1]**, Jorge Víctor Prado-Hernández** [2,*]**, José Luis Silván-Cárdenas** [3]**, Mario Alberto Vázquez-Peña** [4]**, José Manuel Madrigal-Gómez** [3] **and Antonio Martínez-Ruíz** [5]

[1] Programa de Posgrado en Ingeniería Agrícola y Uso Integral del Agua, Universidad Autónoma Chapingo (UACh), Km. 38.5 Carretera México–Texcoco Chapingo, 56230 Texcoco, Estado de México, Mexico; cbolasanchez@gmail.com

[2] Departamento de Suelos, Universidad Autónoma (UACh), Km. 38.5 Carretera México–Texcoco Chapingo, 56230 Texcoco, Estado de México, Mexico

[3] Centro de Investigación en Ciencias de Información Geoespacial (CentroGeo), Contoy 137, Lomas de Padierna, 14240 Tlalpan, Ciudad de México, Mexico; jsilvan@centrogeo.edu.mx (J.L.S.-C.); jmadrigal@centrogeo.edu.mx (J.M.M.-G.)

[4] Departamento de Irrigación, Universidad Autónoma Chapingo, Km. 38.5 Carretera México–Texcoco Chapingo, 56230 Texcoco, Estado de México, Mexico; mvazquezp@chapingo.mx

[5] Instituto Nacional de Investigaciones Forestales, Agrícolas y Pecuarias (INIFAP), Campo Experimental San Martinito, C.P. 74100 Tlahuapan, Puebla, Mexico; amartinezr8393@gmail.com

\* Correspondence: jpradoh@chapingo.mx

**Abstract:** Rainfall interception plays a role in the hydrological cycle and is a critical component of water balances at the basin level, which is why understanding it is very important; as a result, in recent years, various authors have proposed different models to explain this process and identify which of them adapts better to each forest species. In this context, the aim of this research was to evaluate the Gash model and the sparse Gash analytical model in *Pinus hartwegii* Lindl. and *Abies religiosa* (Kunth) Schltdl. et. Cham., using measurements from 20 precipitation events recorded in May and June 2018 at the Zoquiapan Experimental Forest Station, Mexico. The evaporation rate was calculated using the Penman–Monteith method (PM) and Gash's calculation procedure. The canopy parameters were also calculated with two methods, a graphical one (A) and a method proposed in this research (B), which is based on point cloud generated with drone photogrammetry. For *P. hartwegii*, the most accurate model to estimate interception per rainfall event was the Gash model with the A and Gash methods, which were used to calculate the canopy parameters and evaporation rates, respectively; for accumulated interception, the sparse Gash analytical model with the B and PM methods was used. For *A. religiosa*, the best fit for individual events was presented by the sparse Gash analytical model with the B and PM methods, and for accumulated interception, it was the Gash model with the B and Gash methods. The results allow concluding that the B method proposed in this research is a good alternative for the calculation of rainfall interception, since it tends to improve its estimation, shortening the time for acquiring information about the parameters of the canopy structure and thus minimizing the costs involved.

**Keywords:** canopy parameters; drone photogrammetry; Gash model; stemflow; throughfall; rainfall partitioning

## 1. Introduction

Rainfall interception by the canopy is a component of the hydrological cycle and is caused by the retention in the plant cover foliage of a fraction of the incident precipitation, and it influences surface runoff, water infiltration into the soil, and the amount of water evaporated from the canopy, among other processes. Derived from this, it is important to quantify and model the amount of precipitation intercepted to include it in the water

balances at the river-basin scale and to evaluate the effects of the forest canopy on the amount of water resources available [1,2].

Most mathematical models of rainfall interception are based on a canopy water balance and include variables associated with forest structural characteristics and the meteorological conditions of the site [3,4]. Horton [5], at the beginning of the 20th century, was the first to suggest a model based on canopy storage capacity, duration of the storm, evaporation rate and surface, and rainfall intensity. With this, he established the foundations for the later development of more complex and accurate models, which are still in use today and are considered as classical models of rainfall interception [6,7]. The first of these is the Rutter model based on physical principles that implies a strict analysis of the storms and requires diverse variables.

Rutter's conceptual model is based on a dynamic balance of water in the canopy; it considers that the incident precipitation is partitioned into precipitation entering the canopy, precipitation entering the trunk, and translocation-free precipitation. Precipitation entering the canopy can be evaporated or drained if it exceeds the storage capacity of the canopy; a similar situation occurs with precipitation entering the trunk. This model considers that drainage and evaporation rates are dependent on the amount of water stored in the canopy, so they vary throughout each precipitation event [8,9]. Later, in 1979, Gash [10,11] presented the first analytical model, which simplified Rutter's model, classifying rainfall events and reducing the amount of meteorological information required. The classification consists of grouping events according to rainfall magnitude and canopy and trunk capacity. The time intervals between events should be long enough for the canopy and trunk to dry out; also, each precipitation event is divided into three main phases: wetting up, saturation, and drying out. It is assumed that the mean rainfall intensity and average evaporation rate remain constant in all precipitation events. New proposals for improving the results obtained with the classical models emerged later [7,12–16]. Valente et al. [7] presented the sparse version of the Gash and Rutter models, preserving their theoretical foundation and the required input information, but including two important corrections: (1) they separated the canopy-covered area and the area without canopy, and (2) they separated canopy and trunk evaporation, and considered only the former in the canopy balance.

In addition to the classical models previously addressed, there are other techniques to estimate interception such as the numerical [17], stochastic [18], and parametric [4] ones, and recently, new techniques and tools have been developed to facilitate the obtaining of large-scale canopy structure information [19–21]. Carlyle-Moses and Gash [3] suggest including the use of emergent technologies as tools to specify and improve the estimation of rainfall interception, such as remote perception through satellite images, photogrammetry, radar, and LIDAR, which can acquire information on canopy parameters. As a result of this, studies have emerged recently that use this type of technology in some models to calculate rainfall interception. Vegas-Galdos et al. [19] proposed a simple method based on Rutter's model [8,9] and the information obtained by the Moderate Resolution Imaging Spectroradiometer (MODIS). As a result, they obtained a series of maps of canopy rainfall interception in three watersheds in northern Spain and concluded that remotely sensed data are useful for estimating rainfall interception. Cui and Jia [20] also used MODIS and information collected at the experimental site (both on the canopy structure and meteorological variables) to develop the RS-Gash model based on the Gash model [10]; their model yielded good accuracy for both experimental sites, with RMSE values of 0.34 and 0.60 mm d$^{-1}$. Hassan et al. [21] used the Gash analytical model [10] with information collected in the field to estimate rainfall interception in individual trees, and the results obtained were extrapolated to make estimates at a greater scale (basin level) using satellite images. All of the above shows that using remote sensing data for estimating rainfall interception has good potential. Hence, this study proposed the use of drone photogrammetry to generate a point cloud to recreate the structure of the trees and thus obtain the canopy parameters. Field measurements of translocation and cortical flows and data from a weather station were also used to estimate rainfall interception with two analytical models.

Given the large variety of existing models to predict rainfall interception, it is important to identify the appropriate model for specific forest canopy conditions. According to what was reported by Muzylo et al. [22], the most frequently applied model today is the Gash model, which was proposed in 1979 [10] with modifications. Rainfall interception has been estimated in different types of vegetation, such as Mediterranean forests [7,21,23,24], tropical forests [25], coniferous forests [4,6,20], shrubs and herbs [26,27], secondary forests [28,29], deciduous forests [30], and semi-arid vegetation [31].

*P. hartwegii* and *A. religiosa* forests are dominant in the Sierra Nevada, which forms part of the Neovolcanic Belt. They are important because of the environmental services they provide, such as $CO_2$ sequestration, soil retention, protection against soil erosion, microclimate regulation, wildlife habitat protection, and rainwater infiltration into the soil, the last of which could potentially recharge the aquifer. However, the cover of these forests has decreased due to deforestation and land use change to agricultural, livestock, and urban uses, among others. If the loss of forest cover continues, rainfall retention could decrease considerably and groundwater recharge could be affected, which would substantially alter the water balance, reduce water availability, and cause drought in the area. Locally, forests of both species contribute to the water resources of several major population centers in the states of Mexico and Puebla. This shows the need to estimate rainfall interception for *P. hartwegii* and *A. religiosa* in order to calculate a more accurate water balance that will help to formulate proposals for improved water resource management. However, the estimation of rainfall interception through direct on-site measurements has limitations, due to the time required for the quantifications of the flows that compose it, as well as the cost of buying, installing, and operating measuring equipment. Therefore, mathematical modeling can be a viable alternative for estimating rainfall interception, and the option that best suits the characteristics of each species should be sought. Previous studies reported rainfall interception values of 26.1% and 19.2% for *A. religiosa* and *P. hartwegii*, respectively [32]. However, there is no modeling based on canopy characteristics and the meteorological conditions that can predict rainfall interception by these species in future storms. Therefore, this research study had the following objectives: (1) to estimate canopy rainfall interception by *P. hartwegii* and *A. religiosa* using the Gash model and the sparse Gash analytical model; (2) to obtain the canopy parameters of *P. hartwegii* and *A. religiosa*, and the meteorological parameters required for both models; (3) to propose a new method to obtain canopy parameters from the structure of the trees in each experimental plot, using point cloud and drone photogrammetry; and (4) to evaluate the accuracy of the two models used in both forest species, in order to determine the model that best adapts to the characteristics of each forest species.

## 2. Theory

### 2.1. Gash Model (1979)

The model presented by Gash in 1979 [10] conserves objectivity and physical reasoning in its analysis and derivation, similarly to Rutter's model [8,9], and it incorporates some characteristics of the linear regression model of incident precipitation versus rainfall interception for its deduction [7].

Precipitation is represented as a series of discrete events, each one of them with three different phases. One of the phases is wetting up, whose beginning coincides with the start of precipitation and ends when the canopy is saturated. Another one is saturation, which begins when the canopy storage capacity is exceeded. The third is the drying out phase, which begins at the end of precipitation and ends when the tree is fully dried. Among the precipitation events considered in the modeling, there should be a period of drying.

The rainfall interception per set of events or at the daily level, assuming the occurrence of a single event per day, is calculated as

$$\sum_{j=1}^{n+m} I_j = n(1 - p - p_t)P'g + \left(\frac{\overline{E}}{\overline{R}}\right) \sum_{j=1}^{n} (Pg_j - P'g) + (1 - p - p_t) \sum_{j=1}^{m} Pg_j + qS_t + p_t \sum_{j=1}^{m+n-q} Pg_j \quad (1)$$

where $p$ is the free throughfall coefficient (%); $S_t$ the storage capacity of the trunk (mm); $p_t$ the precipitation fraction that is directed to the branches and trunk of the tree; $Pg_j$ is the incident precipitation of the $j$-th rainfall event (mm); $\overline{E}$ is the mean evaporation rate during rainfall (mm h$^{-1}$); $\overline{R}$ is the mean rainfall intensity during saturated canopy conditions (mm h$^{-1}$); and $P'g$ is the amount of precipitation necessary to saturate the canopy (mm). The variable $P'g$ is calculated as

$$P'g = -\frac{\overline{R}S}{\overline{E}}\, ln\left[1 - \frac{\overline{E}}{\overline{R}(1-p-p_t)}\right] \tag{2}$$

where $S$ is the canopy storage capacity under conditions of zero evaporation (mm). The parameters $p$, $p_t$, $S_t$, and $S$ are associated with the canopy and are obtained following the graphical method described in the Rutter model [8,9].

The parameter $\overline{E}$ is calculated using the Penman–Monteith equation [33], and $\overline{R}$ is taken from the average of the meteorological records of rainfall intensity. The theory of Gash indicates that $\overline{E}$ and $\overline{R}$ should be estimated under conditions of saturation; however, in practice, this restriction is difficult to fulfill, which is why Gash suggests considering the saturated canopy when the accumulated precipitation is equal to or higher than 0.5 mm per hour.

The model assumes that the meteorological conditions based on parameters $\overline{E}$ and $\overline{R}$ prevail in all the precipitation events.

The components of interception, throughfall, and stemflow are considered according to the criteria indicated in Table 1.

**Table 1.** Components of rainfall interception for the Gash model.

| Component of the Interception | Formulation | |
|---|---|---|
| Rainfall interception by canopy | | |
| For m events of insufficient precipitation to saturate the canopy ($Pg \leq P'g$) | $(1-p-p_t)\sum_{j=1}^{m} Pg_j$ | |
| For n events of sufficiently large precipitation to saturate the canopy ($Pg \geq P'g$) | $n(1-p-p_t)P'g - nS$ | Wetting up phase |
| | $\left(\frac{\overline{E}}{\overline{R}}\right)\sum_{j=1}^{n}\left(Pg_j - P'g\right)$ | Saturation phase |
| | $nS$ | Drying out phase |
| Rainfall interception by trunk | | |
| For q events of precipitation that manage to saturate the trunk $\left(Pg \geq \frac{S_t}{p_t}\right)$ | $qS_t$ | |
| For m+n−q events of precipitation that do not saturate the trunk $\left(Pg \leq \frac{S_t}{p_t}\right)$ | $p_t\sum_{j=1}^{m+n-q} Pg_j - qS_t$ | |

### 2.2. The Sparse Gash Analytical Model by Valente et al. 1997

Gash et al. [11] reformulated their original model from 1979 with the intention of improving the estimations of rainfall interception for sparse forests, because the first model did not consider this type of condition.

The modified model assumes the division of the study area into an open area and an area covered by canopy (*c*), which improves the accuracy in the estimation of the mean evaporation rate under saturation conditions, since it only includes the evaporation generated by canopy-covered areas, assuming zero evaporation in non-covered areas. The Valente et al. [7] model considers the above in calculating the precipitation required for

canopy saturation ($P'g$) and includes a correction in the evaporation amount involved in the process, since it excludes the evaporation on the trunk from its calculation ($\varepsilon\overline{E}_c$) and considers only the evaporation of the canopy under saturation conditions ($\overline{E}_c$), which is why the $\overline{E}_c$ term from the Gash et al. [11] model is replaced by $(1-\varepsilon)\overline{E}_c$. $P'g$ is recalculated as

$$P'g = -\frac{\overline{R}}{(1-\varepsilon)\overline{E}c}\frac{S}{c}ln\left[1 - \frac{(1-\varepsilon)\overline{E}c}{\overline{R}}\right]. \tag{3}$$

Valente et al. [7] suggested that the drainage from the canopy of the j-th rainfall event ($Dr_j$) was only present in the precipitation events that surpass the saturation. Considering a water balance in the canopy for these precipitation events, the drainage for a rainfall event j was expressed with Equation (4).

$$Dr_j = c\left[1 - \frac{(1-\varepsilon)\overline{E}c}{\overline{R}}\right]\left(Pg_j - P'g\right) \tag{4}$$

It was also suggested to include $P''g$, which represents the amount of precipitation necessary to saturate the trunks (Equation (5)). The introduction of parameter $c$ modifies the formulation of the components of the rainfall interception. It also modified the modeling of the stemflow proposed by Gash et al. [11], including parameter $p_d$, to quantify the percentage of the precipitation that is diverted to the trunk once the canopy was saturated (Table 2).

$$P''g = \frac{\overline{R}}{\overline{R} - (1-\varepsilon)\overline{E}c}\frac{S_t}{p_d c} + P'g \tag{5}$$

**Table 2.** Components of rainfall interception for the sparse Gash analytical model.

| Component of the Interception | Formulation | |
|---|---|---|
| Rainfall interception by canopy | | |
| For m events of insufficient precipitation to saturate the canopy ($Pg \leq P'g$) | $c\sum\limits_{j=1}^{m} Pg_j$ | |
| For n events of precipitation sufficiently large to saturate the canopy ($Pg \geq P'g$) | $c(n\,P'g) - nS$ | Wetting up phase |
| | $c\left\{\left[\frac{(1-\varepsilon)\overline{E}c}{\overline{R}}\right]\sum\limits_{j=1}^{n}\left(Pg_j - P'g\right)\right\}$ | Saturation phase |
| | $cnS$ | Drying out phase |
| Rainfall interception by trunk | | |
| For q events of precipitation that manage to saturate the trunk $\left(Pg \geq P''g\right)$ | $qS_t$ | |
| For n-q events of precipitation that do not saturate the trunk $\left(Pg \leq P''g\right)$ | $p_d\,c\left[1 - \frac{(1-\varepsilon)\overline{E}c}{\overline{R}}\right]\sum\limits_{j=1}^{n}\left(Pg_j - P'g\right)$ | |

## 3. Materials and Methods

### 3.1. Study Site

The study area was at the Zoquiapan Experimental Forest Station located in the municipality of Ixtapaluca, State of Mexico, with elevations of 3200 to 3500 masl and an area of 1638 ha. Annual precipitation ranges from 900 to 1200 mm per year, and the mean annual temperature is 11.1 °C. The predominant species are *Abies religiosa*, *Pinus hartwegii*, and *Alnus firmifolia*.

Two 0.25 ha experimental plots, one of *P. hartwegii* and the other of *A. religiosa,* were established to obtain the rainfall interception data (Figure 1). The leaf area index (LAI) was 2.83 for *P. hartwegii* and 2.99 for *A. religiosa*; this index was estimated with an LI-COR® brand sensor, model LAI-2200C. For this purpose, four profiles inside each plot were made, measuring 0.5 m each: two diagonally, one in a north–south direction, and the other in an east–west direction. The *P. hartwegii* trees had a mean Diameter at Breast Height of 37.56 cm and a mean height of 21.14 m, while the *A. religiosa* trees had a mean Diameter at Breast Height of 50.50 cm and a mean height of 23.40 m.

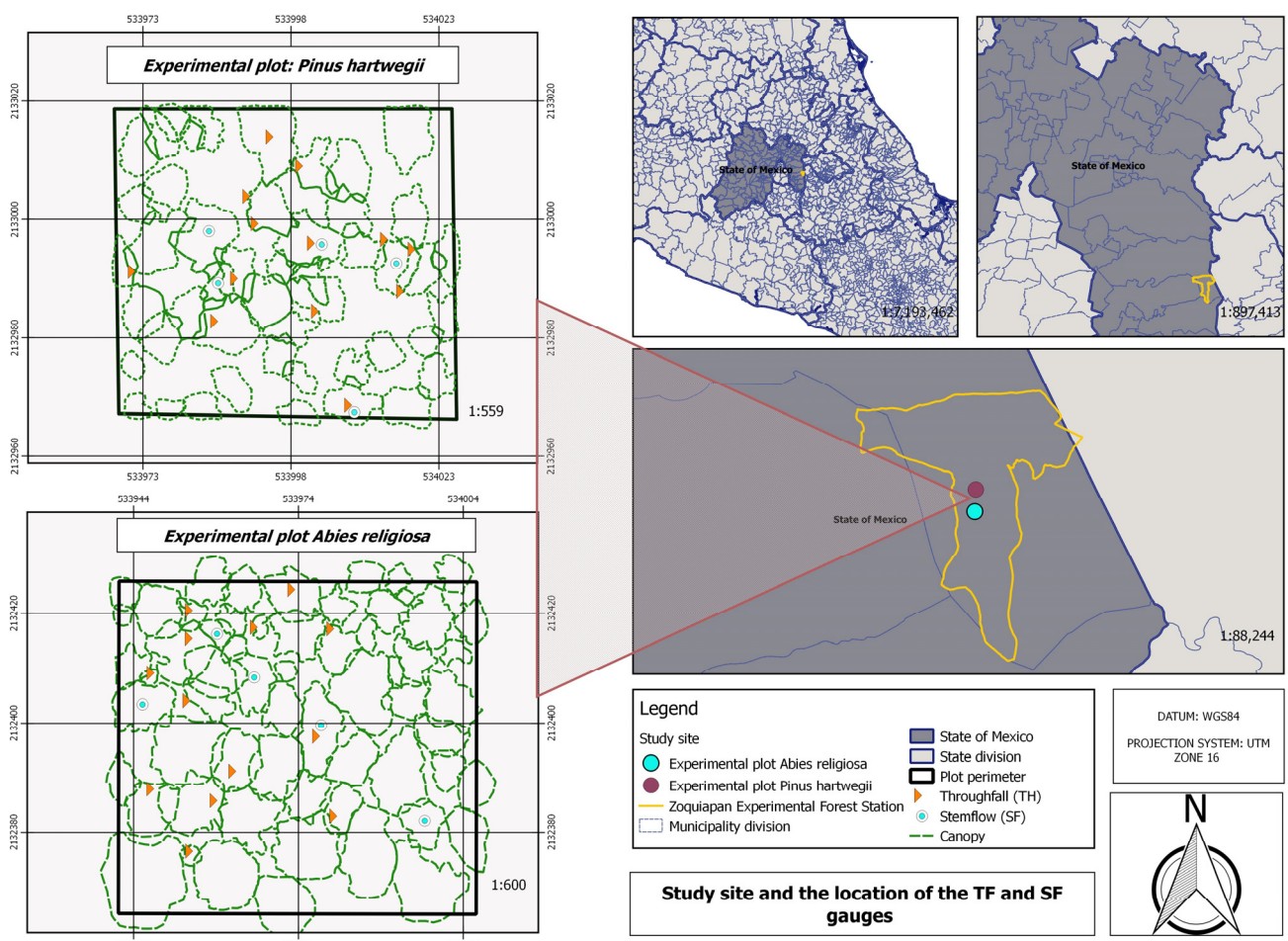

**Figure 1.** Study site and location of instrumentation to measure throughfall and stemflow.

### 3.2. Instrumentation

The measurements were carried out during May and June 2018. The incident precipitation ($Pg$, mm), mean, maximum, and minimum temperature (T, Tmax, Tmin, °C), wind speed (u, ms$^{-1}$), relative humidity (RH, %), solar radiation (Rs, Wm$^{-2}$), and barometric pressure (Pb, mb) were recorded every 10 min by a DAVIS® Cordless Vantage Pro2™ model weather station, located in an open area without canopy, at a distance of approximately 200 to 750 m from the experimental plots. The throughfall (TH, mm) was quantified by using 39 randomly placed collectors per experimental plot, measuring manually with graduated cylinders the rainfall depth recorded by the weather station at the end of each precipitation event (Figure 1). Incident precipitation in the canopy was considered. The stemflow (SF, mm) was measured using five hose collars per plot, which were placed around the trunks of the selected trees, also measuring manually with graduated cylinders. Rainfall interception (I, mm) was estimated per event; for the purposes of this study, one

event per day was considered preferable, and when two or more events occurred in the same day, there should be a separation of at least 6 h between them [19].

Aerial photographs of the experimental plots' vegetation cover were obtained with Phantom 4 and Inspire drones, both DJI® brand. Photographs were taken in June 2018 during four scheduled flights at altitudes of 80 and 100 m, using frontal and lateral overlaps of 90% and 85%, respectively (Table 3). Camera settings were established with respect to site illumination. Several control points were established in both plots using a high-precision Global Positioning System (GPS) to ensure accuracy in assigning the geographic coordinates of the photographed sites.

**Table 3.** Flight configuration.

| Drone | Flight | Plot | Altitude (m) | Overlap | |
|---|---|---|---|---|---|
| | | | | % | Type |
| Inspire | 1 | P. hartwegii | 100 | 90 | Frontal |
| | | | | 85 | Lateral |
| | 2 | | 100 | 90 | Frontal |
| | | | | 85 | Lateral |
| | 3 | A. religiosa | 100 | 90 | Frontal |
| | | | | 85 | Lateral |
| Phantom 4 | 4 | | 80 | 90 | Frontal |
| | | | | 85 | Lateral |

### 3.3. Meteorological Parameters

The calculation of the evaporation rate under saturated canopy conditions was done by two methods. The first used the Penman–Monteith equation [8,9,24,34]:

$$E_o = \frac{\Delta(Rn - G) + \rho Cp \frac{(es-ea)}{r_a}}{\lambda(\Delta + \gamma)} \tag{6}$$

where $E_o$ is the reference evaporation rate (mm h$^{-1}$), $\Delta$ is the slope of the vapor pressure curve (Pa K$^{-1}$), $Rn$ is the net radiation (W m$^{-2}$), $G$ is the soil heat flux (W m$^{-2}$), $\rho$ is the mean air density at constant pressure (kg m$^{-3}$), $Cp$ is the specific heat of the air (J kg$^{-1}$ K$^{-1}$), $(es-ea)$ is the vapor pressure deficit (Pa), $r_a$ is the aerodynamic resistance (m h$^{-1}$), $\lambda$ is the latent vaporization heat (J kg$^{-1}$), and $\gamma$ is the psychometric constant (Pa K$^{-1}$). The value of 0.15 was used for the albedo in both species, as suggested by Klingaman et al. [30] and Villalobos and Fereres [35]. For the evaporation calculation, we considered the data from the weather station corresponding to the duration periods of the precipitation events that had a recorded depth greater than or equal to 0.5 mm in one hour (Table 4).

The aerodynamic resistance $r_a$ is estimated as [25,36]

$$r_a = \frac{\left(ln \frac{Z-d}{Z_0}\right)^2}{k^2 u} \tag{7}$$

where $k$ is the Von Karman constant (0.41), $Z$ is the height at which wind speed was measured (m), $d$ is the zero plane displacement height (m), $Z_0$ represents the roughness length of the surface, and $u$ is the wind speed (m h$^{-1}$). To obtain the values of $Z_0$ and $d$, they were calculated as 0.1 h and 0.75 h, respectively, where h is the average height of trees [8,9,36,37].

**Table 4.** Parameters and constants used in the calculation of evaporation $E_o$.

| Symbol | Parameter | Value | Units |
|--------|-----------|-------|-------|
| $\rho$ | Air density at constant pressure | 1.05 | kg m$^{-3}$ |
| $Cp$ | Specific heat of the air | 1013.00 | J kg$^{-1}$ K$^{-1}$ |
| $\gamma$ | Psychometric constant | 66.00 | Pa K$^{-1}$ |
| $\lambda$ | Latent heat of vaporization | 2.45 | J kg$^{-1}$ |
| $A$ | Albedo | 0.15 | n/a |

The mean evaporation rate obtained with the Penman–Monteith (PM) equation for saturated canopy conditions was calculated by averaging the values of all precipitation events recorded under saturation conditions.

Gash [10] assumes that the slope of the regression between rainfall interception ($I$) and precipitation depth per event ($Pg$) is equal to the $\overline{E}/\overline{R}$ quotient of Rutter's model (Equations (8) and (9)), assuming that the mean evaporation rate ($\overline{E}$) and mean rainfall intensity ($\overline{R}$) are constant over all precipitation events. The Gash assumption holds for events that satisfy the canopy saturation criterion, which implies that the hourly accumulated precipitation must be greater than 0.5 mm. In this study, the slope proposed by Gash [10] was obtained for *P. hartwegii* and *A. religiosa* (Figure 2a,b). The mean evaporation rate ($\overline{E}$) was calculated with the $\overline{E}/\overline{R}$ quotient obtained and the mean rainfall intensity ($\overline{R}$) recorded with the weather station used in this study, similarly to Hassan et al. [21]. This evaporation was called the mean Gash evaporation rate.

$$I = a\,Pg + b \tag{8}$$

$$I = \left(\frac{\overline{E}}{\overline{R}}\right)Pg + \left[S + \int_0^{t'} \overline{E}dt\,(1 - \left(\frac{\overline{E}}{\overline{R}}\right)1 - p - p_t)\right] \tag{9}$$

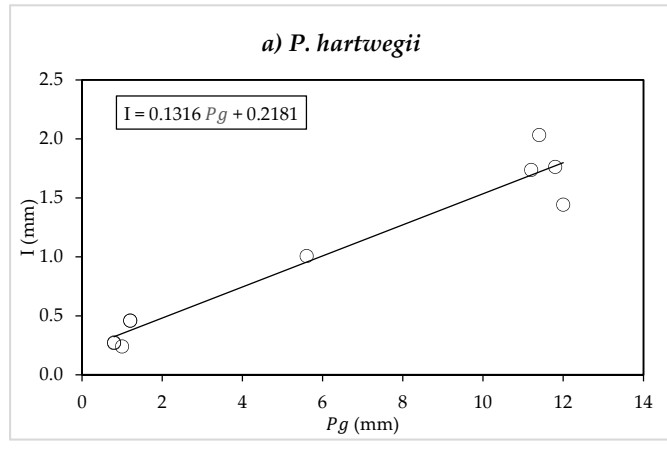

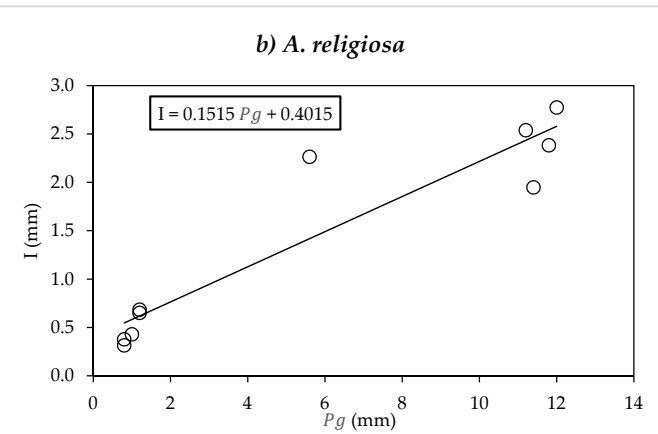

**Figure 2.** Relationship between the incident precipitation and rainfall interception of the events that met the canopy saturation condition, proposed by Gash, for (**a**) *P. hartwegii* and (**b**) *A. religiosa*.

### 3.4. Parameters of the Canopy Structure

The parameters derived from the canopy for the Gash model were $S$, $p$, $S_t$, and $p_t$. For the sparse Gash analytical model, the parameters $S_c$, pd, and $c$ were obtained. Two methods were implemented in the estimation of these parameters: a graphical method (A), based on field measurements and using linear regressions, as described by Rutter et al. [8,9];

and a point cloud method (B), proposed in this study, which is based on the values of the LAI and drone photogrammetry.

### 3.4.1. Method A

In the graphical method, a linear regression was performed between the translocation flow (TH) and incident precipitation ($Pg$) (Figure 3a,b) of the events that met the canopy saturation condition proposed by Gash in order to obtain the S and $p$ values. At the intersection point of the fit line with the horizontal axis, the TH component is null, and it is therefore assumed that the incident precipitation ($Pg_{int}$) corresponds to the water amount stored in the canopy (S) and the evaporation from the canopy (E). The value of S was obtained by subtracting from $Pg_{int}$ the value of the component $\overline{E}$, which in this study is considered equal to the Penman–Monteith evaporation ($E_0$). The parameter $p$ was taken to be the slope of the fit line between TH and $Pg$. The reasoning is based on a simple balance in the canopy proposed by Rutter et al. [8]:

$$Pg - TH = S + \overline{E}. \tag{10}$$

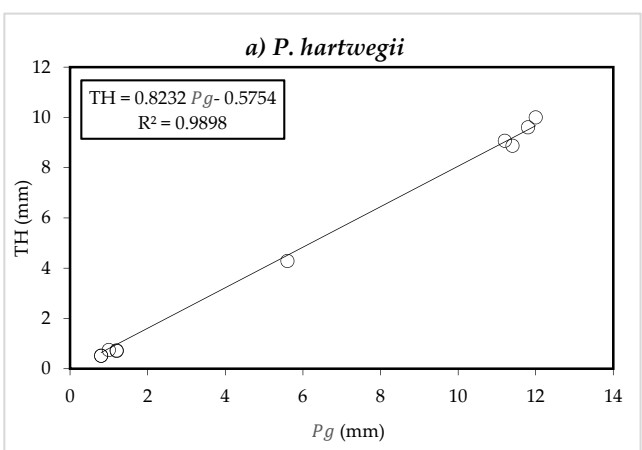
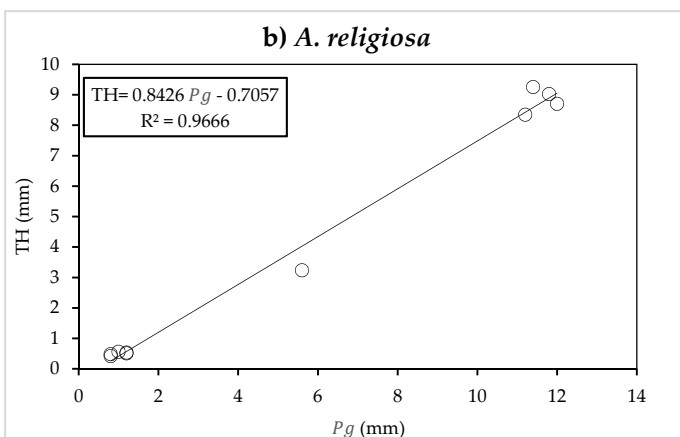

**Figure 3.** Relationship between the translocation flow and incident precipitation of the events that met the canopy saturation condition, as proposed by Gash, for (**a**) *P. hartwegii* and (**b**) *A. religiosa*.

The parameters $S_t$ and $p_t$ were determined with a procedure similar to that applied to obtain $S$ and $p$, based on a linear regression between SF and $Pg$, excluding evaporation from the canopy in the calculations. In the sparse Gash analytical model, $S_c$ was calculated by the quotient of $S$ divided by $c$, and pd assumed the value of $p_t$. The value of $c$ was obtained with the B method, which is explained in the following section.

### 3.4.2. Method B

In this method, $S$ was calculated with the expression $S = 0.3LAI$, as suggested by Deguchi et al. [28]. $S_t$ was obtained from the product of $S$ multiplied by the constant $\epsilon$, as suggested by Valente et al. [7], and $S_c$ resulted from dividing $S$ by $c$. Drone photogrammetry was used to estimate the parameters $p$, $p_t$, $p_d$, and $c$. The parameter $c$ corresponded to the percentage of cover of the orthogonal projection of the canopy over the ground; the orthoimages of the experimental plots were used for this purpose. This value of $c$ was also used for the graphical method (A).

The number of photographs obtained for the *P. hartwegii* plot was 207 and for the *A. religiosa* plot, it was 130. The photographs were processed in three sequential stages using Pix4D Mapper Pro software Version 4.5.6 (Pix4D SA, Lausanne, Switzerland): in stage one (initial processing), a calibration with control points was made; in stage two (point cloud and mesh and DSM), the point cloud was generated and finally, in stage three (orthomosaic and index), 3D models and an orthomosaic were constructed.

### 3.4.3. Fractioning of the Point Cloud into Three Classes

For the estimation of parameters $p$, $p_t$, and $p_d$, the point cloud was fractioned into three classes: high green vegetation (canopy), high dry vegetation (trunks and branches), and low vegetation and soils. The procedure for the calculation of the fraction of classes was made in three sequenced stages (Figure 4), where *VI* is the Vegetation Index and *H* is the vegetation height.

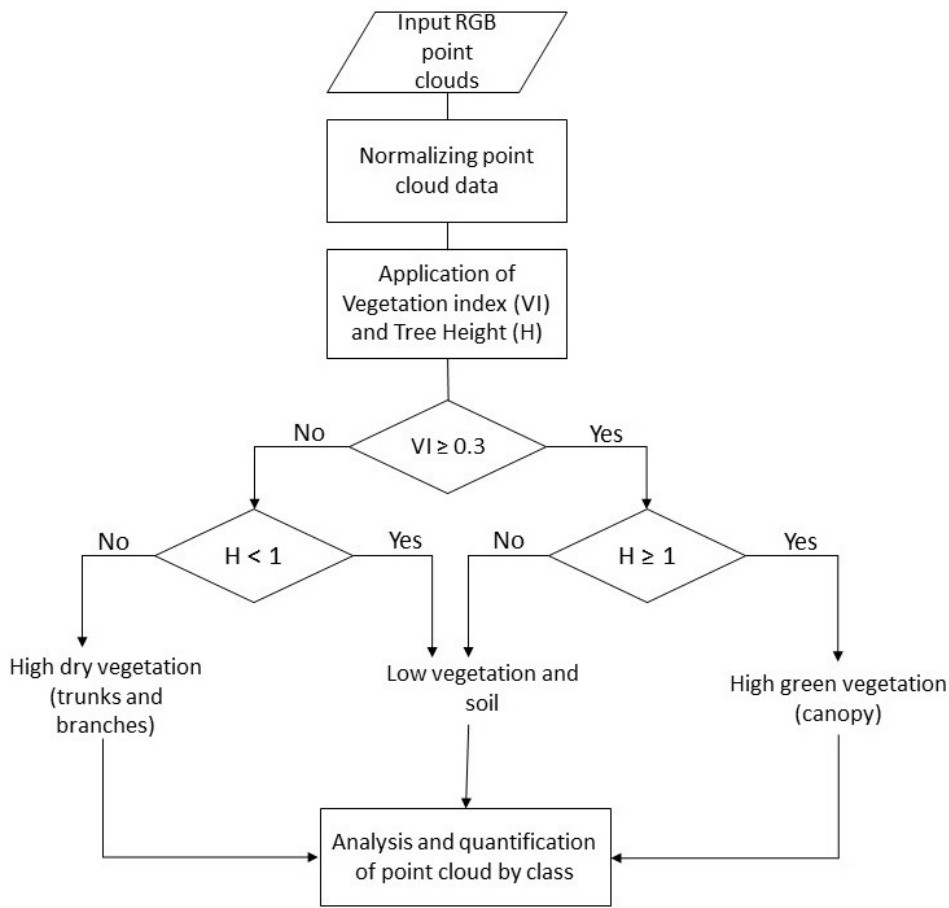

**Figure 4.** Flow diagram of the method proposed for the fractioning of the point cloud per classes.

The first stage of the process, called normalizing point cloud data, had the objective of calculating the heights of the trees and identifying the relief of the soil surface, eliminating the abrupt variability of the elevations. In this process, a rasterization of the point cloud was done using elevation as the value for each pixel, and later, it was segmented and a process of filtration and interpolation was applied [38], in order to finally obtain the normalization of the point cloud from which a layer of soil and a layer of trees were obtained (Figure 5).

In the second stage, the discretization of each point was obtained according to its color, taking as reference the orthoimage of each experimental plot and applying the vegetation index *(VI)* (Equation (11)), as suggested by Silván (2018) [39,40]:

$$VI = \frac{4G - R - B}{4G + R + B} \tag{11}$$

where *G* is the reflectance value of the green band; *R* is the reflectance value of the red band; and *B* represents the reflectance value of the blue band. The index values range from $-1$ to $+1$.

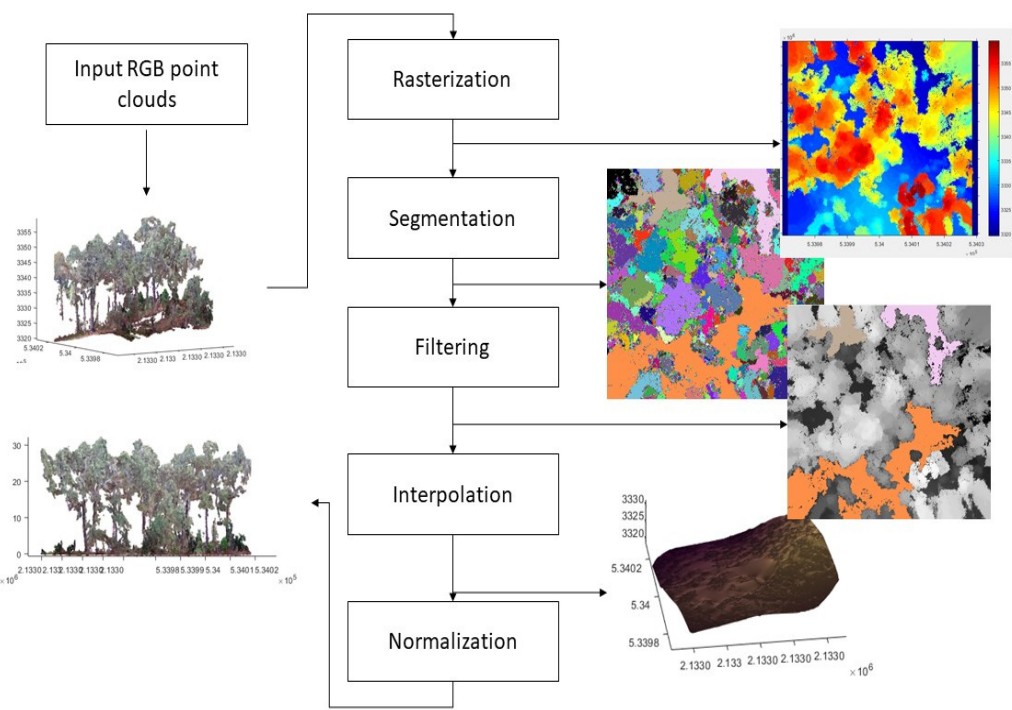

**Figure 5.** Flow diagram for the normalization of the point cloud [39].

Application of the vegetation index *VI* was done in diverse areas of the orthoimage, making the regions with values higher than or equal to 0.3 correspond with vegetation. This criterion of identification of the vegetation agrees with those applied in other studies [41].

The third stage had the objective of differentiating the foliage, trunks and branches, and soil. To differentiate the foliage, a *VI* higher than or equal to 0.3 was used, along with a height (H) greater than or equal to 1 m. To differentiate the trunks and branches, a *VI* lower than 0.3 was used, along with a height greater than or equal to 1 m. Finally, to differentiate the soil, a *VI* lower than 0.3 was used, along with a height greater than or equal to 0.3 and a height lower than 1 m.

The fractions corresponding to the high green vegetation (canopy) and high dry vegetation (branches and trunks) were assigned to parameters *p* and *pt*, respectively. The parameter *pd* was made equivalent to the parameter $p_t$. Parameter *c* was estimated with the classification of color in the orthoimage, using the percentage of area covered by green vegetation and the percentage covered by soil in each plot.

### 3.5. Implementation of the Models

The total number of precipitation events was randomly divided into two groups. Group one was used to obtain the parameters from the models, and group two was used to validate the models. With each group, four simulations were made using the two models chosen, the two forest species, combining at the same time the parameters of the canopy structure with methods A and B, and the evaporation rates obtained with the Penman–Monteith (PM) and Gash (Gash) methods. Thus, eight estimations of rainfall interception were made per forest species: G-(A, PM), G-(A, Gash), G-(B, PM), G-(B, Gash), V-(A, PM), V-(A, Gash), V-(B, PM), and V-(B, Gash), where G and V represent the Gash model and the sparse Gash analytical model, respectively, and what is found in parentheses refers to the method to obtain the canopy parameters and evaporation rate.

*3.6. Validation of the Models*

Evaluation of the accuracy of the models was applied to the group of precipitation events for validation and was made using the root mean square error (*RMSE*) and the Nash–Sutcliffe efficiency coefficient (*NSE*):

$$RMSE = \sqrt{\frac{\sum_{j=1}^{n}\left(I_{m,i} - I_{mod,i}\right)^2}{n}} \tag{12}$$

$$NSE = 1 - \frac{\sum_{i=1}^{n}\left(I_{m,i} - I_{mod,i}\right)}{\sum_{i=1}^{n}\left(I_{m,i} - \overline{I}_{mod}\right)} \tag{13}$$

where $I_{m,i}$ is the interception measured from the rainfall event $i$ (mm); $I_{mod,i}$ represents the modeled interception of the rainfall event $i$ (mm); and $\overline{I}_{mod}$ is the average of interceptions observed or measured from the number of events $n$.

According to the NSE value, the performance of the models can be established as follows [42]: unsatisfactory (NSE < 0.65), acceptable ($0.65 \leq$ NSE < 0.80), good ($0.80 \leq$ NSE < 0.90), and very good (NSE $\geq$ 0.90).

A sensitivity analysis was made of the rainfall interception to the meteorological parameters ($\overline{E}$ and $\overline{R}$) and those associated with the canopy structure ($S$, $p$, $S_t$, $p_t$, $S_c$, and $p_d$), with the aim of identifying those of greatest influence on the results. For this, simulations were made of the response from the two models with modifications to the parameter values of $-30\%$, $-20\%$, $-10\%$, $+10\%$, $+20\%$, and $+30\%$.

## 4. Results and Discussion

*4.1. Precipitation Events*

Twenty precipitation events that took place in May and June 2018 were analyzed. These events had a total depth of 121.60 mm. Ten events were used to obtain the parameters of the models (Table 5), and the rest were used to evaluate the performance of the models (Table 6). The precipitation events used for the parameterization were those that met the canopy saturation conditions established by Gash (precipitation accumulated per hour greater than 0.5 mm).

**Table 5.** Meteorological conditions of the group of precipitation events for the parametrization of the models.

| No. E | Date | $Pg$ [1] (mm) | R [2] (mm h$^{-1}$) | T min [3] (°C) | T max [4] (°C) | Mean T [5] (°C) | RH [6] (%) | BP [7] (mb) | U [8] (m/s) | Rs [9] (W m$^{-2}$) |
|---|---|---|---|---|---|---|---|---|---|---|
| 1 | 19 May 2018 | 11.80 | 3.47 | 7.60 | 16.20 | 8.40 | 94.20 | 1007.68 | 1.92 | 18.76 |
| 3 | 21 May 2018 | 1.20 | 1.80 | 9.20 | 11.40 | 10.24 | 88.00 | 1009.94 | 1.28 | 73.80 |
| 5 | 06 June 2018 | 11.20 | 3.73 | 8.20 | 15.90 | 9.60 | 90.36 | 1005.32 | 2.18 | 28.52 |
| 6 | 07 June 2018 | 11.40 | 17.10 | 8.10 | 14.90 | 10.20 | 85.20 | 1002.44 | 5.78 | 18.40 |
| 8 | 13 June 2018 | 1.20 | 1.02 | 8.20 | 9.20 | 11.00 | 91.00 | 1012.56 | 2.00 | 231.81 |
| 9 | 13 June 2018 | 5.60 | 0.52 | 10.30 | 11.40 | 8.40 | 97.04 | 1011.62 | 0.00 | 4.66 |
| 11 | 15 June 2018 | 12.00 | 2.25 | 9.20 | 12.40 | 9.70 | 96.61 | 1008.22 | 1.21 | 16.20 |
| 15 | 21 2018 | 0.80 | 0.11 | 4.10 | 8.20 | 6.04 | 95.00 | 1011.69 | 0.60 | 1.30 |
| 17 | 22 June2018 | 0.80 | 1.60 | 8.10 | 8.40 | 8.20 | 93.20 | 1012.96 | 0.00 | 0.00 |
| 19 | 24 June2018 | 1.00 | 1.20 | 8.70 | 9.90 | 9.70 | 89.80 | 1011.34 | 0.60 | 90.40 |

[1] Incident precipitation (*Pg*), [2] mean rainfall intensity (R), [3] minimum temperature (T min), [4] maximum temperature (T max), [5] mean temperature (Mean T), [6] relative humidity (RH), [7] barometric pressure (BP), [8] wind speed (u), and [9] solar radiation (Rs).

**Table 6.** Meteorological conditions of the group of precipitation events for the validation of the models.

| No. E | Date | $Pg$ [1] (mm) | R [2] (mm h$^{-1}$) | T min [3] (°C) | T max [4] (°C) | Mean T [5] (°C) | RH [6] (%) | BP [7] (mb) | U [8] (m/s) | Rs [9] (W m$^{-2}$) |
|---|---|---|---|---|---|---|---|---|---|---|
| 2 | 20 May 2018 | 11.40 | 3.34 | 3.40 | 14.70 | 6.60 | 94.97 | 1005.80 | 0.36 | 11.44 |
| 4 | 22 May 2018 | 2.60 | 2.05 | 7.80 | 11.20 | 8.90 | 90.00 | 1010.04 | 2.08 | 0.00 |
| 7 | 12 June 2018 | 6.80 | 0.94 | 7.40 | 10.60 | 8.43 | 95.34 | 1010.59 | 0.38 | 18.17 |
| 10 | 14 June2018 | 17.80 | 1.25 | 7.60 | 10.50 | 9.27 | 96.52 | 1010.79 | 0.11 | 20.46 |
| 12 | 16 June 2018 | 3.60 | 0.51 | 6.80 | 10.00 | 8.80 | 96.72 | 1009.39 | 0.70 | 0.00 |
| 13 | 17 June 2018 | 16.00 | 3.00 | 10.00 | 13.60 | 11.60 | 92.72 | 1008.47 | 3.00 | 227.54 |
| 14 | 18 June 2018 | 1.20 | 1.03 | 9.20 | 10.10 | 9.67 | 98.00 | 1010.76 | 2.20 | 118.25 |
| 16 | 22 June2018 | 0.40 | 0.34 | 7.80 | 8.10 | 7.80 | 95.42 | 1013.55 | 3.60 | 0.00 |
| 18 | 23 June 2018 | 0.40 | 0.21 | 8.40 | 8.80 | 8.50 | 93.09 | 1012.66 | 0.00 | 0.00 |
| 20 | 25 June 2018 | 4.40 | 0.61 | 7.30 | 11.30 | 8.70 | 94.06 | 1012.76 | 0.40 | 21.09 |

[1] Incident precipitation ($Pg$), [2] mean rainfall intensity (R), [3] minimum temperature (T min), [4] maximum temperature (T max), [5] mean temperature (Mean T), [6] relative humidity (RH), [7] barometric pressure (BP), [8] wind speed (u), and [9] solar radiation (Rs).

### 4.2. Rainfall Interception, Throughfall, and Stemflow

The throughfall (TH) obtained for both species coincides with those found in previous studies in morphologically similar species [2,4,37,43–46]. The stemflow (SF) was small for both species and represented less than 4% of the total rainfall interception (Table 7); this result is lower than what was reported by Llorens and Domingo [45].

**Table 7.** Quantification of the interception components from the 20 rainfall events.

| Species | Throughfall | | Stemflow | | Interception | |
|---|---|---|---|---|---|---|
| | Mm | % | mm | % | mm | % |
| *P. hartwegii* | 97.45 | 73.38 | 4.51 | 3.13 | 19.63 | 23.48 |
| *A. religiosa* | 88.77 | 60.61 | 3.54 | 1.89 | 29.27 | 37.49 |

The rainfall interception estimated for *P. hartwegii* and *A. religiosa* was 23.48% and 37.49%, respectively. The values found are within the ranges reported in studies with similar species [2,29,37,43].

### 4.3. Meteorological Parameters

The mean rainfall intensity ($\overline{R}$) of the precipitation events used to parametrize each model under saturation conditions resulted in 3.28 mm h$^{-1}$.

The mean evaporation rate ($\overline{E}$) estimated with PM was 0.037 mm h$^{-1}$. This parameter was generalized for both species as recommended by Ghimire et al. [29], who suggest that the evaporation from different species in the same area does not vary significantly. The mean evaporation rates obtained by Gash were 0.43 and 0.49 mm h$^{-1}$ for *P. hartwegii* and *A. religiosa*, respectively. The values obtained with PM were approximately ten times lower than the values estimated with Gash in the two forest species. Similar studies also reported differences between the values obtained with PM and Gash [4,21,25,29,30].

In this study, the possible causes of discrepancy between the values obtained with PM and Gash are primarily associated with three aspects: (1) time of occurrence of the rainfall events; (2) uncertainty in the correct application of aerodynamic resistance of the vegetation over the topography of the study zone [21,29]; and (3) the difficulty in quantifying the influence of the density and structure of the canopy in the evaporation speed of raindrops when hitting the vegetation. Regarding the first, the precipitation events took place mostly during the evening–night in absence of radiant energy ($R_n$), which caused a considerable decrease in the estimated values of evaporation during the

storms using Penman–Monteith [30], since the equation gives a higher weight to solar radiation as the main source of energy for the estimation of evaporation. However, despite the low or null values of solar radiation in each storm, there are other climate factors that allow performing the change of liquid precipitation on the vegetation to water vapor, such as air temperature, relative humidity during and after the storm, and wind speed [28,36]. When it comes to the second aspect, there is the possibility of the aerodynamic resistance of the species under study not having the same behavior as the one set out by Thom [36], who studied an ecosystem of scarce vegetation with isolated trees. The third aspect refers to the size of the raindrops and the characteristics of their impact on the vegetation influencing the speed of evaporation [14].

### 4.4. Parameters of the Canopy Structure

The estimated values for the parameters derived from the canopy obtained from methods A and B are summarized in Table 8.

**Table 8.** Estimated values of the canopy structure parameters.

| | *A. religiosa* | | *P. hartwegii* | |
|---|---|---|---|---|
| | **Method A** | **Method B** | **Method A** | **Method B** |
| $S$ | 0.80 | 0.89 | 0.70 | 0.85 |
| $p$ | 0.84 | 0.51 | 0.82 | 0.67 |
| $S_t$ | 0.03 | 0.02 | 0.45 | 0.19 |
| $p_t$ | 0.03 | 0.20 | 0.04 | 0.16 |
| $c$ | 0.83 | 0.83 | 0.70 | 0.70 |

The canopy storage capacity ($S$) of both forest species was within the range of 0.3 to 3.0 mm reported for conifer forests [46,47], and it was also within the range reported in forest species with similar characteristics to the two species studied whose values fall between 0.4 and 2.0 mm [2,7,29,44]. In this study, the canopy storage capacity in *A. religiosa* was higher than in *P. hartwegii*, in the two methods employed for its calculation (A and B). The differences are associated with the morphological characteristics of each species, as well as the leaf area index value reported, since according to Zhang and Li [27] and Ghimire et al. [29], canopy storage capacity is linearly related to this index and depends on the specific storage of the canopy.

The free throughfall coefficient $p$ for *P. hartwegii* was higher than the one estimated for *A. religiosa* with method B, and it was the opposite with method A. The values estimated with method A were approximately 13 to 15% higher than those calculated with the point cloud of method B, and their values are high compared to the 0.13–0.62 range obtained in similar studies [2,4–6]. The estimations of the graphical method (A) could be affected by the shock effect of the raindrops on the canopy when it was saturated [29]. On the other hand, the values of $p$ calculated from the fraction of the point cloud corresponding to high green vegetation (canopy) assume better estimations, because they were derived from the canopy geometry from photographs taken with drones in the study plots. The values of $p$ obtained with method B were similar to the 0.53 reported by Fan et al. [4] for *Pinus elliotti* and *Pinus caribaea* forests.

The canopy storage capacity of the trunk $St$ and the fraction of precipitation diverted to the trunk $pt$ were small for both species with the two calculation methods employed. Usually, these parameters are much lower than those related to the canopy, and there are no accurate reference values, since they are also influenced by qualitative characteristics of the trunk and branches. Since its values are low, their impact is not relevant with regard to other parameters in the interception process, but it is suggested that they are considered to improve the accuracy in the estimations [29].

The value of parameter *c*, which corresponds to the soil fraction that is covered by the canopy of trees in each study plot, was higher in *A. religiosa*. The fraction of voids in the canopies was not considered in this study, and it could be considered as a possible source of error.

### 4.5. Performance of the Models Considered

In *P. hartwegii*, the G-(A, Gash) model was the most accurate in the estimation of interception per event, with an RMSE value of 0.37 mm and an NSE value of 0.72 (Table 9). This is because the evaporation rate used results from the same model and therefore offers a better approximation [21]. The next best models in terms of accuracy were V-(A, PM) and G-(A, PM) with negligible differences between them; their RMSE and NSE values were 0.39 mm and 0.69, and 0.39 and 0.68, respectively.

**Table 9.** Values of the interception observed ($I_m$) and modeled ($I_{mod}$) for *P. hartwegii* with the combinations of the meteorological and canopy parameters.

| No. E | Date | Pg (mm) | I obs (mm) | $I_{mod}$ of the Gash Model (mm) | | | | $I_{mod}$ of the Sparse Gash Analytical Model (mm) | | | |
|---|---|---|---|---|---|---|---|---|---|---|---|
| | | | | (A [1], PM [3]) | (A, Gash [4]) | (B [2], PM) | (B, Gash) | (A, PM) | (A, Gash) | (B, PM) | (B, Gash) |
| 2 | 20 May 2018 | 11.40 | 1.68 | 1.25 | 2.05 | 1.14 | 2.10 | 1.27 | 2.54 | 1.16 | 2.42 |
| 4 | 22 May 2018 | 2.60 | 0.71 | 0.47 | 0.47 | 1.05 | 0.66 | 0.77 | 1.00 | 0.97 | 1.17 |
| 7 | 12 June 2018 | 6.80 | 1.17 | 1.02 | 1.22 | 1.09 | 1.41 | 0.93 | 1.63 | 1.11 | 1.83 |
| 10 | 14 June 2018 | 17.80 | 1.84 | 1.32 | 2.72 | 1.21 | 2.94 | 1.34 | 3.36 | 1.23 | 3.24 |
| 12 | 16 June 2018 | 3.60 | 0.55 | 0.65 | 0.65 | 1.38 | 0.84 | 0.81 | 1.15 | 1.08 | 1.42 |
| 13 | 17 June 2018 | 16.00 | 2.21 | 1.30 | 2.68 | 1.19 | 2.70 | 1.32 | 3.13 | 1.22 | 3.01 |
| 14 | 18 June 2018 | 1.20 | 0.47 | 0.22 | 0.22 | 0.58 | 0.40 | 0.71 | 0.79 | 0.84 | 0.84 |
| 16 | 22 June 2018 | 0.40 | 0.11 | 0.07 | 0.07 | 0.19 | 0.13 | 0.26 | 0.27 | 0.22 | 0.23 |
| 18 | 23 June 2018 | 0.40 | 0.13 | 0.07 | 0.07 | 0.19 | 0.13 | 0.26 | 0.27 | 0.22 | 0.23 |
| 20 | 25 June 2018 | 4.40 | 1.08 | 0.79 | 0.79 | 1.64 | 0.98 | 0.84 | 1.27 | 1.09 | 1.53 |
| ∑ Interception (mm) | | | 9.95 | 7.16 | 10.94 | 9.67 | 12.30 | 8.51 | 15.41 | 9.15 | 15.93 |
| RMSE (mm) | | | | 0.39 | 0.37 | 0.54 | 0.42 | 0.39 | 0.69 | 0.46 | 0.70 |
| NSE | | | | 0.68 | 0.72 | 0.40 | 0.63 | 0.69 | 0.01 | 0.55 | −0.03 |

[1] A: Obtaining canopy parameters with the graphical model, [2] B: Obtaining canopy parameters with the method proposed by point cloud, [3] PM: Mean evaporation rate with Penman–Monteith method, [4] Gash: Mean evaporation rate with Gash. Example: in the combination (A, PM), the graphical method was used to calculate the canopy parameters and the Penman–Monteith method was used to calculate the evaporation.

It is important to highlight that the three most accurate models were the ones that used the canopy parameters obtained with the graphical method (A). However, the method proposed in this study (B) also had good results, since the G-(B, Gash) and V-(B, PM) models had slightly lower accuracy than the V-(A, PM) and G-(A, PM) models; in them, RMSE was from 0.42 to 0.46 mm, and NSE was from 0.63 to 0.55, respectively.

The least accurate models were the combinations of V-(A, Gash) and V-(B, Gash), whose results overestimated the measured interceptions. RMSE was 0.69 and 0.70 mm, and NSE was 0.01 and −0.03, respectively.

The models that had a better approximation to the total accumulated interception for the twenty precipitation events were the combinations V-(B, PM), V-(A, PM), and G-(A, Gash), with relative errors of −6.51%, −9.75%, and 8.64%, respectively (Figure 6a). As a consequence, the accumulated estimations are satisfactory because generally, the estimation that is mostly used is the interception accumulated by periods of time that include more than one rainfall event [2,19,21,29]; however, this type of estimation can be overvalued due to the compensation of errors of the set of events so that an analysis per rainfall event will allow more reliable results.

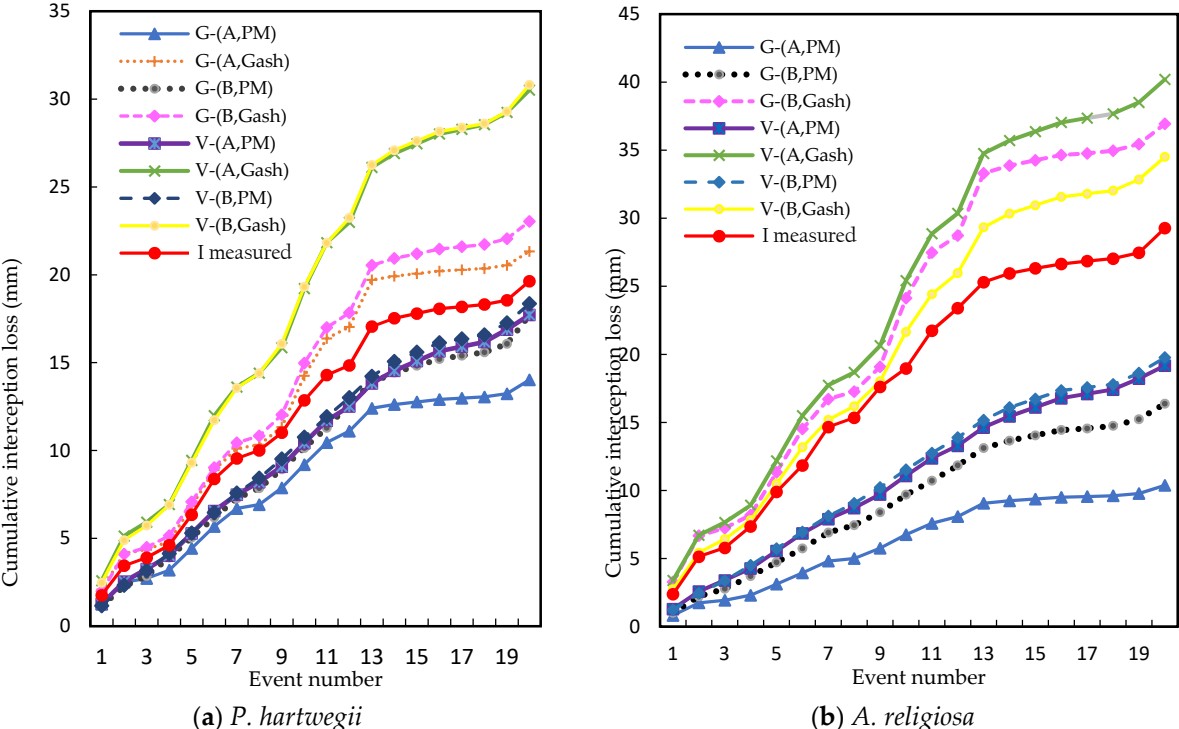

**Figure 6.** Cumulative measured and simulated rainfall interception for both calibration and validation groups for *P. hartwegii* (**a**) and for *A. religiosa* (**b**).

In *A. religiosa*, the estimations of the interception per individual precipitation event were not as satisfactory as in *P. hartwegii*; for this forest species, the most accurate model was V-(B, PM), with an RMSE of 0.61 mm and an NSE of 0.52 (Tables 9 and 10). This result is similar to the one reported by Pérez et al. [2] for *Pinus pinea*. For the estimation of total accumulated interception, the combinations of G-(B, Gash) and V-(B, Gash) were more accurate, with relative errors of 17.86% and 26.17%, respectively (Figure 6b). In this forest species, it was not possible to model the interception with the G-(A, Gash) combination because results from $P'g$ without physical meaning were obtained when the component $(1\text{-}p\text{-}p_t)\,\overline{R}$ was lower than the mean evaporation rate ($\overline{E}$), generating with it a logarithm with a negative value that does not exist by definition [10]. This was due to the incompatibility of the graphical method (A) in conjunction with the estimation of the evaporation rate by the Gash method (Gash) for dense species such as the ones used in this study.

Method B contributed with good results in the estimation of rainfall interception in both species. This method has the following advantages over the graphical method A: (i) it does not require prior experimental data on throughfall and stemflow for the estimation of canopy parameters; (ii) estimation of the parameters related to the canopy structure through the treatment of aerial photographs taken by low-cost drones; (iii) acceleration in obtaining information; (iv) decrease in costs for information acquisition; and (v) decrease in field work. Given this, and based on the results, it is considered that method B, as proposed

in this study, is a viable alternative to feed estimation models of rainfall interception when there is no experimental information.

**Table 10.** Values of the interception observed ($I_m$) and modeled ($I_{mod}$) for *A. religiosa* with the combinations of the meteorological and canopy parameters.

| No. E | Date | $Pg$ (mm) | $I_m$ (mm) | $I_{mod}$ of the Gash Model (mm) | | | $I_{mod}$ of the Sparse Gash Analytical Model (mm) | | | |
|---|---|---|---|---|---|---|---|---|---|---|
| | | | | (A [1], PM) | (B [2], PM [3]) | (B, Gash [4]) | (A, PM) | (A, Gash) | (B, PM) | (B, Gash) |
| 2 | 20 May 2018 | 11.40 | 2.76 | 0.92 | 1.22 | 2.79 | 1.29 | 2.76 | 1.82 | 2.69 |
| 4 | 22 May 2018 | 2.60 | 1.57 | 0.37 | 0.94 | 0.96 | 0.87 | 1.15 | 1.13 | 1.40 |
| 7 | 12 June 2018 | 6.80 | 2.82 | 0.87 | 1.17 | 1.96 | 1.05 | 1.87 | 1.85 | 2.02 |
| 10 | 14 June 2018 | 17.80 | 1.36 | 1.00 | 1.29 | 3.94 | 1.36 | 3.01 | 1.30 | 3.27 |
| 12 | 16 June 2018 | 3.60 | 1.65 | 0.50 | 1.13 | 1.25 | 0.92 | 1.32 | 1.14 | 1.55 |
| 13 | 17 June 2018 | 16.00 | 1.91 | 0.97 | 1.27 | 3.62 | 1.34 | 2.85 | 1.28 | 3.15 |
| 14 | 18 June 2018 | 1.20 | 0.65 | 0.19 | 0.54 | 0.56 | 0.82 | 0.91 | 0.93 | 1.00 |
| 16 | 22 June 2018 | 0.40 | 0.22 | 0.06 | 0.20 | 0.20 | 0.32 | 0.32 | 0.22 | 0.23 |
| 18 | 23 June 2018 | 0.40 | 0.18 | 0.06 | 0.20 | 0.20 | 0.32 | 0.32 | 0.22 | 0.23 |
| 20 | 25 June 2018 | 4.40 | 1.81 | 0.60 | 1.14 | 1.48 | 0.95 | 1.45 | 1.15 | 1.67 |
| $\sum$ Interception [5] (mm) | | | 14.92 | 5.55 | 9.08 | 16.96 | 9.23 | 15.95 | 11.04 | 17.19 |
| RMSE (mm) | | | | 1.12 | 0.82 | 1.05 | 0.86 | 0.71 | 0.61 | 0.78 |
| NSE | | | | −0.63 | 0.14 | −0.42 | 0.04 | 0.35 | 0.52 | 0.22 |

[1] A: Obtaining canopy parameters with the graphical model, [2] B: Obtaining canopy parameters with the method proposed by point cloud, [3] PM: Mean evaporation rate with the Penman–Monteith method, [4] Gash: Mean evaporation rate with Gash. Example: in the combination (A, PM), the graphical method was used to calculate the canopy parameters, and the Penman–Monteith method was used to calculate the evaporation, [5] Interception: sumatoria de los eventos individuales observados y modelados.

The sparse Gash analytical model was better with the use of the mean evaporation rate ($\overline{E}$) calculated with the Penman–Monteith method in relation to the two forest species studied, with a more favorable effect in *P. hartwegii* (Table 9). In the Gash model, neither of the two methods used to calculate $\overline{E}$ showed a clear favorable trend; in *P. hartwegii*, good results were obtained with the Gash method (Table 9), but with this method, an unfavorable behavior was observed in *A. religiosa*, since the G-(B, Gash) combination resulted in one of the highest values of RMSE (1.05 mm) and the G-(A, Gash) combination could not be executed (Table 10).

### 4.6. Sensitivity Analysis

The results show that both the Gash model and the sparse Gash analytical model are sensitive to parameters $\overline{E}$, $\overline{R}$, and $S$. In this study, with an increase of 30% in $\overline{E}$ and $S$, the estimated interception increased 11.90% and 6.54%, respectively. On the other hand, an increase of the same magnitude in $\overline{R}$ generated a decrease of 10.32% in the interception calculated; the negative slope in this parameter means that with higher rainfall intensity, the canopy saturates more quickly and, as consequence, it intercepts less precipitation (Figure 7).

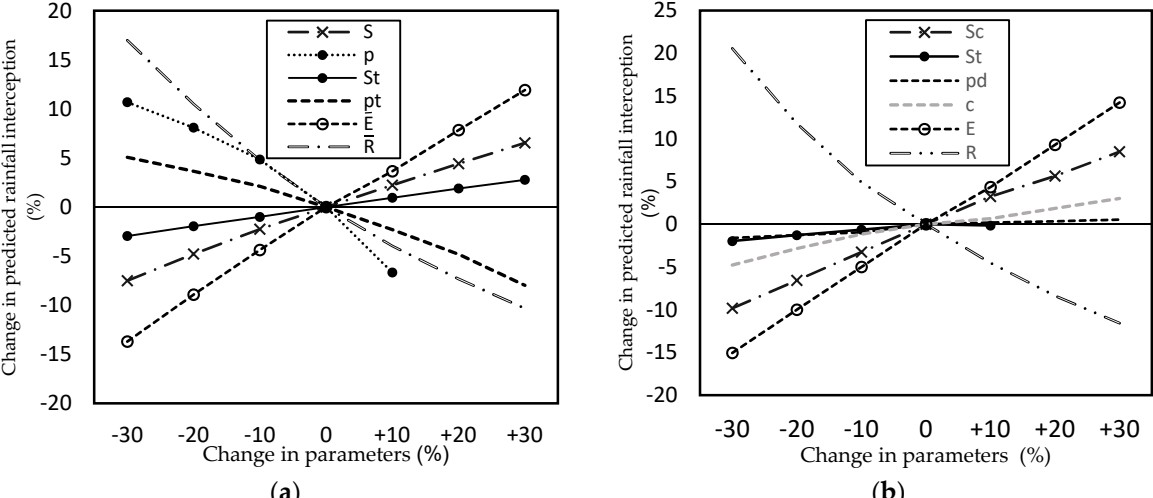

**Figure 7.** Sensitivity analyses of canopy parameters ($S$, $S_c$, $S_t$, $p$, $p_t$, $p_d$, and $c$) and meteorological parameters ($\overline{E}$ and $\overline{R}$) of (**a**) Gash model and (**b**) Sparse Gash analytical model.

The least sensitive parameter for both models was $S_t$; with an increase of 30% in its value, the estimated rainfall interception increased 2.7%. In the sparse Gash analytical model, the parameter with the lowest sensitivity was $p_d$; with an increase of 30% in its value, there was only a 0.55% increase in estimated rainfall interception. These results coincide with those obtained by Cui and Jia [20], Fan et al. [4], Limousin et al. [13], Sun et al. [47], Valente et al. [7], and Zhang and Li [27].

The parameter $p_t$ is usually considered as having low sensitivity [4,20,27]. In fact, a decrease of 30% in its value in this research produced an increase of 5.07% in estimated rainfall interception, which could be considered as largely irrelevant.

In the sparse Gash analytical model, the parameter $c$ is generally reported as highly sensitive and with similar effects as $\overline{E}$, $\overline{R}$ and $S$ [4,24]. However, in this study, parameter $c$, although it did not show high sensitivity, was also not insignificant, and its inclusion in the model is important because it is responsible for fractioning the real area that corresponds to the canopy cover.

The high sensitivity of rainfall interception to parameters $\overline{E}$, $\overline{R}$, and $S$ suggests continuing to research methodologies to improve the estimation of those parameters. For the calculation of parameter $\overline{E}$ with the Penman–Monteith method, the following should be considered: amount of radiant energy ($R_n$) during the occurrence of the events, aerodynamic resistance of the species under study [37], and effect on the vegetation of the size of the raindrops [14]. Instead of using a single value of $\overline{R}$, which is representative for the rainfall regime of a site, it would be advisable to obtain values for the set of events with similar rainfall characteristics, for example, in events of low, moderate, and high rainfall intensity. An alternative to specify the value of parameter $S$ could be through LIDAR [48,49] images; however, the accuracy–cost relation regarding the use of drone photography would have to be analyzed.

## 5. Conclusions

Method B, proposed in this study to obtain the parameters of the canopy structure, is an alternative with good potential to estimate the rainfall intercepted with the Gash model and sparse Gash analytical model in the species *P. hartwegii* and *A. religiosa*.

For *P. hartwegii*, the sparse Gash analytical model, with method B and Penman–Monteith to estimate the evaporation, was the most accurate to estimate the total interception accumulated with a relative error of −6.51%. For individual events, the Gash model with method A to calculate the canopy parameters and with the evaporation rate of the Gash method was the most accurate with an RMSE of 0.37 mm and an NSE of 0.72.

In the case of *A. religiosa*, the best fit for accumulated interception was presented by the Gash model, with method B and with the estimated evaporation rate by the Gash method, with a relative error of 17.86%. For individual events, the most accurate model was the sparse Gash analytical model, with method B and with the evaporation rate of Penman–Monteith, with an RMSE of 0.61 mm and an NSE of 0.52.

The fit of the forecasted intercepted depths by the models analyzed to those observed was more accurate per rainfall event than for the accumulated depth for all events taken together.

The sparse Gash analytical model had a better performance with the use of the mean evaporation rate calculated by the Penman–Monteith method in both forest species. In the Gash model, a clear trend was not observed in favor of any of the two methods used to estimate the mean evaporation rate.

The two models analyzed were highly sensitive to the parameters $\overline{E}$, $\overline{R}$, and S, and with low sensitivity to $S_t$.

**Author Contributions:** Conceptualization, C.B.-S. and J.V.P.-H.; methodology, C.B.-S., J.V.P.-H., J.L.S.-C., M.A.V.-P. and J.M.M.-G.; software, J.L.S.-C.; validation, C.B.-S., J.V.P.-H., J.L.S.-C. and M.A.V.-P.; formal analysis, C.B.-S. and J.V.P.-H.; sensitivity analysis, C.B.-S., J.V.P.-H. and A.M.-R.; investigation and resources, C.B.-S., J.V.P.-H., J.L.S.-C., M.A.V.-P. and J.M.M.-G.; data curation, C.B.-S., J.V.P.-H., J.L.S.-C. and J.M.M.-G.; writing—original draft preparation, C.B.-S. and J.V.P.-H.; writing—review and editing, C.B.-S., J.V.P.-H., J.L.S.-C. and A.M.-R.; visualization, supervision, and project administration, C.B.-S., J.V.P.-H., J.L.S.-C., M.A.V.-P. and J.M.M.-G. All authors have read and agreed to the published version of the manuscript.

**Funding:** This research was funded by University of Chapingo and National Council of Science and Technology (CONACyT) of Mexico.

**Data Availability Statement:** The data presented in this study are available on request from the corresponding author.

**Acknowledgments:** The authors thank the staff of the EFEZ for their support in the realization of the experimental phase and to the Centro de Investigación en Ciencias de Información Geoespacial A. C. "CentroGeo" for their invaluable technical collaboration.

**Conflicts of Interest:** The authors declare no conflict of interest.

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
