# Peer review of "Estimating Rainfall Interception of Pinus hartwegii and Abies religiosa Using Analytical Models and Point Cloud"

_forests, doi:10.3390/f12070866_

Round 1

Reviewer 1 Report

This is a well organized and thoughtful study of rainfall interception modeling of two species in Mexico.  The use of remote sensing enhances the novelty of this study and would be of interest to scientists studying ecohydrological processes.  I would recommend that a native English speaker read the manuscript for syntax errors.

As an example, see the first independent clause of the first sentence in the Introduction ("Rainfall interception is the hydrological process where forest canopy fraction of the incident precipitation,...").  This sentence should be rewritten for clarity, as well as some others. 

Author Response

We appreciate this assessment, also shared by the second reviewer. We have rewritten several ideas and revised the English language writing throughout the paper so that the ideas put forward are better understood.

Some the modifications in the manuscript have been highlighted in blue.

Reviewer 2 Report

This manuscript focuses on estimating rainfall interception of Pinus hartwegii and Abies religiosa. The authors introduce a new approach to estimate the canopy parameters (required by the Gash model), based on point clouds generated from drone photogrammetry. As the authors correctly stated, rainfall interception by vegetation is an important water balance component, thus studies like this provide significant information on ecohydrological processes. However, it is difficult to understand the meaning of some sentences. Thus, extensive editing of English language is required. More comments/suggestions below:

Abstract

  • Why did the authors used both the original and the revised versions of the Gash model?

If they wanted to compare the difference between fully-covered and sparse forest, they could easily set the canopy cover fraction (c) in the revised model to 1. This, in theory, should be the same as the original Gash model.

Introduction

  • It is difficult to “extract” the meaning of most of the sentences. Please check again and rephrase.

  • Better to give few more details about some widely used models (Rutter, Gash original and revised). Which are the parameters used in the Gash model? How do we derive them? What is the meaning of these parameters? What did other studies found? What is the novelty of this study?

  • What is the reasoning/justification for objective 1?

  • Objective 2. “to obtain the canopy and meteorological parameters”. Of what? Please write it out

  • Objective 3. What do the authors mean “from its architecture”. Please rephrase this sentence.

  • Objective 4. Check my comment above about the use of the two versions of the Gash model.

Section 2

  • It is very difficult to read this section. Please revise.

  • What is the meaning of the symbol (1- ε) in equations 3 - 5 ?

Sections 3.1 - 3.2

  • A map and few figures of the study site and the location of the TF gauges would be very informative.

  • How do the authors define the “precipitation event”: 3 hours separation interval? 6 hours separation interval? Daily?

  • Lines 173 – 174. Photographs of what? The authors should expand further on this method since it is the novelty of this dtudy.

Section 3.3

  • Lines 176 – 180. Please check the descriptions of the parameters. Evaporation is computed with the Penman – Monteith equation. The mean evaporation rate during saturated condition is computed from this equation but as the average for all hours where P > 0.5 mm h-1 (lines 197 – 199). Better to use the symbol Eo for the Penman – Monteith equation.

  • Lines 200 – 210. Any references about this method? Also, can you provide more details? Other authors use the “mean” method (Klaasen et al., 1998) which allows the derivation of all the parameters needed for the Gash model (E/R, c, S) with the use of two regression lines (before saturation and after saturation). There are few other regression methods available (Jackson 1975, Gash and Morton 1978, Leyton et al. 1967).

Section 3.4.1

  • Better to add a figure (regression line) showing how the parameters were derived with the regression method.

Section 3.5.1

  • I think that this section should be in Results.

  • Line 282. 6 hours separation ? Or daily separation (Table 4 and 5)?

  • Table 4. N.6. How is it possible to have a mean rainfall intensity higher than the total rainfall?

Section 4.3

  • Table 7. c is equal to 1-p. However, in this table the numbers do not match. Maybe I missed something? Also, I don’t understand why c is blank in method A. According to the authors for this method “Parameter c was obtained from orthoimages, represented the percentage of coverage of canopies per each experimental plot.”

Section 4.4

  • Are the results from tables 8 and 9 the same as in figure 3?

  • According to the authors:

“The mean rainfall intensity (?Ì…) of the precipitation events used to parametrize each model under conditions of saturation resulted in 3.28 mm h -1.

The mean evaporation rate (?Ì…) estimated with PM was 0.037 mm h-1.

The mean evaporation rates obtained by Gash were 0.43- and 0.49-mm h-1 for P. hartwegii and A. religiosa, respectively.”

Also, S in table 7 is between 0.7 and 0.8. Based on the parameter values above, I would expect lower I in some of the results given in tables 8 and 9. The differences in I between the original and the revised Gash models are higher than expected. Could you elaborate more on these results?

Author Response

Point-by-point responses to all comments received are provided in an attached file.

Round 2

Reviewer 2 Report

I would like to congratulate the authors for their effort in improving the manuscript and in responding to all of my comments. I believe the manuscript is now ready for publication.